# Interleukin-17, a salivary biomarker for COVID-19 severity

**Fatemeh Saheb Sharif-Askari[1], Narjes Saheb Sharif-Askari[1], Shirin Hafezi[1], Bushra Mdkhana[1], Hawra Ali Hussain Alsayed[2], Abdul Wahid Ansari[3], Bassam Mahboub[4], Adel M. Zakeri[5], Mohamad-Hani Temsah[6], Walid Zahir[7,8], Qutayba Hamid[1,9,10], Rabih Halwani**  [1,9,11] *

**1** Sharjah Institute of Medical Research, University of Sharjah, Sharjah, United Arab Emirates, **2** Pharmacy Department, Dubai Health Authority, Dubai, United Arab Emirates, **3** Dermatology Institute, Translational Research Institute, Hamad Medical Corporation, Doha, Qatar, **4** Rashid Hospital, Dubai Health Authority, Dubai, United Arab Emirates, **5** Department of Plant Production, Faculty of Agriculture and Food Sciences, King Saud University, Riyadh, Saudi Arabia, **6** Department of Pediatrics, Immunology Research Lab, College of Medicine, King Saud University, Riyadh, Saudi Arabia, **7** G42 Health Care, Abu Dhabi, United Arab Emirates, **8** Institute of Public Health, United Arab Emirates University, Al Ain, United Arab Emirates, **9** Department of Clinical Sciences, College of Medicine, University of Sharjah, Sharjah, United Arab Emirates, **10** Meakins-Christie Laboratories, Research Institute of the McGill University Health Center, Montreal, Quebec, Canada, **11** Prince Abdullah Ben Khaled Celiac Disease Chair, Department of Pediatrics, Faculty of Medicine, King Saud University, Riyadh, Saudi Arabia

* rhalwani@sharjah.ac.ae

**Data Availability Statement:** If the data are all contained within the manuscript and/or Supporting information files, enter the following: All relevant data are within the manuscript and its Supporting information files.

## Abstract

### Objectives

T-helper 17 cell-mediated response and their effector IL-17 cytokine induced by severe acute respiratory syndrome coronavirus 2 (SARS-CoV-2) infection is a major cause of COVID-19 disease severity and death. Therefore, the study aimed to determine if IL-17 level in saliva mirrors its circulatory level and hence can be used as a non-invasive bio-marker for disease severity.

### Methods

Interleukin-17 (IL-17) level was evaluated by ELISA in saliva and blood of 201 adult COVID-19 patients with different levels of severity. The IL-17 saliva level was also associated with COVID-19 disease severity, and need for mechanical ventilation and/or death within 29 days after admission of severe COVID-19 patients.

### Results

We found that IL-17 level in saliva of COVID-19 patients reflected its circulatory level. High IL-17 level in saliva was associated with COVID-19 severity (P<0.001), need for mechanical ventilation (P = 0.002), and/or death by 29 days (P = 0.002), after adjusting for patients' demographics, comorbidity, and COVID-19 serum severity markers such as D-Dimer, C-reactive protein, and ferritin.

**Funding:** This research has been financially supported by Tissue Injury and Repair (TIR) group operational grant (Grant code: 150317); COVID-19 research grant (CoV19-0307); Seed grant (Grant code: 2001090275); and by collaborative research grant (Grant code: 2001090278) to RH, University of Sharjah, UAE; and by a Sandooq Al Watan Applied Research & Development grant to RH (SWARD-S20-007); and by Prince Abdullah Ben Khalid Celiac Disease Research Chair, under the Vice Deanship of Research Chairs, King Saud University, Riyadh, Kingdom of Saudi Arabia. Moreover, the funders had no role in study design, data collection and analysis, decision to publish, or preparation of the manuscript.

**Competing interests:** The authors have declared that no competing interests exist.

## Conclusion

We propose that saliva IL-17 level could be used as a biomarker to identify patients at risk of developing severe COVID-19.

## Introduction

The number of cases and deaths due to coronavirus disease 2019 (COVID-19) are still continuing to increase [1]. COVID-19 pneumonia can progress to acute lung injury (ALI) and acute respiratory distress syndrome (ARDS) secondary to overwhelming inflammatory responses [2–4]. The current management of COVID-19 is supportive, and therefore, it is suggested that all patients with severe COVID-19 should be screened for hyper-inflammation or "cytokine storm" in order to identify those who would benefit from targeted immunosuppression or immunomodulation to prevent ALI/ARDS [5]. Currently, there is no specific marker to distinguish, at an early stage COVID-19, patients who are prone to progression of COVID-19 disease from others.

Among the many cytokines involved in the cytokine storm, interleukin-17 (IL-17) is a notable and predominant mediator of pulmonary inflammation [6]. Dysregulation of T helper 17 (Th-17) cells and enhanced expression of IL-17 in the lungs promote the production of downstream pro-inflammatory molecules such as interleukin-1beta (IL-1β), TNF alpha (TNFα), interleukin-6 (IL-6), neutrophil chemoattractants such as interleukin-8 (IL-8), and monocyte chemoattractant protein-1 (MCP-1/CCL2). Recruited neutrophils then induce reactive oxygen species, leading to ALI and protein-rich inflammatory lung infiltration, the hallmark features of ARDS [6, 7]. Consistently, increased IL-17 level in mice with lipopolysaccharides (LPS)-induce acute lung injury was associated with greater infiltration of inflammatory cells to the lung and decreased overall survival [8]. Furthermore, addition of exogenous IL-17 further exacerbated LPS-induced production of TNFα, IL-1β, and IL-6, revealing the pathogenic role of IL-17 as a key upstream modulator of the inflammatory pathway. In the same study, mice genetically deficient in IL-17 or those that received anti-IL-17 antibodies had a better survival, less lung infiltration and better lung pathology scores following LPS challenge [8]. IL-17 was also shown to synergy TNFα and IL-1β via the mitogen-activated protein kinase (MAPK) pathways [9], which is known to be activated by different groups of viruses [10] and by SARS-CoV-2 [11].

Following SARS-CoV-2 infection, the severity of disease was shown to positively correlate with plasma levels of IL-1β, IL-6, TNFα, interferon gamma (IFNγ), and IL-17A proinflammatory cytokines [2, 11–15]. Several reports have also associated the increased IL-17A levels and Th17 response in upper and lower respiratory tracts of COVID-19 patients with COVID-19 severity. In addition, we and others have shown that the level of several plasma biomarkers can be successfully reflected in saliva [16, 17]. Therefore, we hypothesized that salivary IL-17A level may mirror its plasma level and hence can be used as a non-invasive biomarker for disease severity. We evaluated IL-17A, TNFα, IL-1β protein levels in saliva of COVID-19 patients with different severities, and found that among these cytokines, IL-17A was associated with COVID-19 severity and poor patient survival outcomes.

## Materials and methods

### Ethics statement

Ethical approval was obtained from the Dubai Scientific Research Ethics Committee (DSREC-08/2021_14). Written, informed consents were obtained from all study participants prior to inclusion.

## COVID-19 patients' cohort

The cohort consisted of 201 adult patients with PCR-confirmed SARS-CoV-2 infection who were referred to Rashid Hospital in Dubai between May 28 and June 30, 2020. Out of 201 COVID-19 patients, 67 patients were asymptomatic, 81 patients had mild to moderate symptoms, and 53 patients had severe disease. Samples were collected on diagnosis of COVID-19 from non-hospitalized asymptomatic or those with mild symptoms, and at admission to hospital from hospitalized patients. Clinical and laboratory data were all collected from these patients at the time of samples collection (Table 1). The COVID-19 severity status was defined as COVID-19 pneumonia requiring high-flow oxygen therapy [18]. Patients with severe COVID-19 were followed up for 29 days after the date of hospital admission. In the samples collected from 201 patients, ELISA (enzyme-linked immunosorbent assay) were used to measure level of IL-17A and two other inflammatory cytokines known to contribute to COVID-19 related pathogenic inflammation—TNFα and IL-1β—and assessed their level with severity and patient survival [19, 20]. IL-17A will be referred to as IL-17 in the rest of this study. As references, and to serve as controls, the level of these cytokines were measured in saliva of 50 healthy controls. The precautions recommended by CDC for safe collecting, handling and testing of biological fluids were strictly followed [21].

## Collection of saliva

As previously reported [22], we followed the unstimulated whole saliva collection method. Before saliva collection we asked participants to sit upright with their head slightly titled downward allowing saliva to collect on the floor of the mouth. The first sample was discarded to eliminate the unwanted debris. The subsequent saliva sample (around 2 mL) was then dribbled into a pre-labeled polypropylene sterile tube. For each participant, salivary flow rate was calculated by dividing the total saliva volume (mL) by the time of collection (min) [23]. The volume

**Table 1. Clinical parameters of COVID-19 patients in according to disease severity.**

| Variables | Healthy controls (n = 50) | COVID-19 Patients | | | P-value |
| | | Asymptomatic (n = 67) | Mild/moderate (n = 81) | Severe (n = 53) | |
|---|---|---|---|---|---|
| Age (years, median, range) | 29 (24–32) | 33 (28–36) | 48 (40–56) | 57 (48–65) | <0.001 |
| Male sex | 31 | 47 | 66 | 44 | 0.030 |
| BMI (median, range) | 24 (22–26) | 25 (22–28) | 27 (24–31) | 28 (26–31) | 0.019 |
| Salivary flow rate | 0.44 ± 0.10 | 0.42 ± 0.18 | 0.39 ± 0.21 | 0.37 ± 0.19 | 0.318 |
| **Comorbidity** | | | | | |
| DM (n,%) | - | 2 (3) | 39 (48) | 27 (54) | <0.001 |
| **Serum severity markers** | | | | | |
| D-dimer (0–0.5 μ/mL) | - | 0.27 (0.18–1.29) | 0.71 (0.38–1.65) | 1.35 (1.04–6.74) | <0.001 |
| CRP (1.0–3.0 mg/L) | - | 1.25 (0.40–7.9) | 18.8 (3–98.3) | 81.3 (23.2–141.6) | 0.003 |
| Ferritin (10–204 ng/mL) | - | 45.2 (37–75) | 535 (234–1197) | 886 (465.8–1612.4) | 0.002 |
| **Cytokines values*** | | | | | |
| Plasma IL-17, pg.mL$^{-1}$ | 20.8 (19–22) | 28 (25–30) | 28.8 (26–31) | 63.4 (51–75) | <0.001 |
| Saliva IL-17, pg.mL$^{-1}$ | 50 (48–51) | 71.9 (66–77) | 78.8 (73–84) | 138.8 (128–149) | <0.001 |
| Saliva TNFα, pg.mL$^{-1}$ | 170.3 (158–182) | 463.9 (402–525) | 506.9 (454–558) | 568 (507–628) | <0.001 |
| Saliva IL-1β, pg.mL$^{-1}$ | 30 (26–34) | 44.8 (39–50) | 61.4 (55–68) | 71.6 (64–79) | <0.001 |

Abbreviation: BMI, body mass index; CRP, C-reactive protein. Detection limits for ELISA assay of IL-17 is 15.6 pg.mL$^{-1}$, TNFα is 15.63 pg.mL$^{-1}$, and IL-1β is 3.91 pg. mL$^{-1}$.

*Unadjusted P-values.

of saliva was determined by weighing, considering a density of 1 g/mL for saliva. All samples were then stored at –20˚C until immediately before use.

## ELISA assays of IL-17, IL-1β, and TNFα cytokines

IL-17, IL-1β and TNFα cytokine concentrations were determined in saliva and/or plasma samples using commercially available human ELISA kits (Human IL-17, DY317-05, R&D; Human IL-1β, DY201-05, R&D; and human TNFα ELISA KIT, ab181421, Abcam). For the assays, saliva samples were centrifuged at 700g for 15 minutes at 4˚C, and the supernatant was used. Diluent optimization was performed for the saliva samples. We performed assays following the manufacturers' instructions. All samples were measured in duplicates.

## Gene expression data sets

The gene expression data sets of COVID-19 nasopharyngeal swabs (GSE152075) [24], including 430 patients with SARS-CoV-2 infection and 54 uninfected individuals, and COVID-19 lung autopsies (GSE150316) [25], including 52 COVID-19 fatal cases and 5 SARS-CoV-2-uninfected individuals, were all publicly available at the National Center for Biotechnology Information Gene Expression Omnibus (NCBI GEO, http://www.ncbi.nlm.nih.gov/geo) or the European Bioinformatics Institute (EMBL-EBI, https://www.ebi.ac.uk).

## Analysis procedures

Association of IL-17, IL-1β, or TNFα saliva concentrations and COVID-19 severity were evaluated using regression models adjusted for patients' demographic factors including age, male gender, and body mass index (BMI); comorbidities such as diabetes mellitus (DM); and serum markers of COVID-19 severity such as D-dimer, C-reactive protein (CRP), and ferritin. Association of IL-17, IL-1β, or TNFα saliva concentrations of severe COVID-19 patients and the need for mechanical ventilation and/or death within 29 days from admission was evaluated using Cox proportional hazards regression models adjusted for all the above-mentioned patient demographics, comorbidities, and markers of COVID-19 severity. Kaplan–Meier survival curves were then constructed to show cumulative survival over the 29 days period. All selected variables in the models were tested for the presence of collinearity by evaluating variance inflation factors and magnitude of standard errors. Furthermore, the discriminatory power of models was assessed using the area under the curve (AUC). Discrimination refers to the ability of a model to clearly distinguish between 2 groups of outcomes (discriminate between severe and non-severe patients with COVID-19) and can range from 0.5 (no discrimination) to 1.0 (perfect discrimination) [26].

Moreover, for the bioinformatic analysis, the data was pre-processed using the Bioconductor package *limma-voom* [27]. The fold change of differential expressed genes were carried out using *Limma* Bioconductor package [28, 29].

For evaluating IL-17 mRNA levels in whole blood of COVID-19 patients we have used the following primers: human IL-17A, forward, 5'-3': CGGACTGTGATGGTCAACCTGA, and reverse, 5'-3': GCACTTTGCCTCCCAGATCACA; human 18s, forward, 5'-3': TGACT-CAACACGGGAAACC, and reverse, 5'-3': TCGCTCCACCAACTAAGAAC. Gene expression was analyzed using the Comparative Ct (ΔΔCt) method after normalization to the housekeeping gene 18 s rRNA. Analysis was performed using R software (v 3.0.2), SPSS Version 26 (IBM Corporation, Chicago, USA), and Graphpad Prism 8 (GraphPad Software Inc., San Diego, USA). All tests were two-tailed and a P value of less than 0.05 was considered statistically significant. A file consisting of all patient's parameters used in the analysis is provided in S1 File.

## Results

### Cohort characteristics

Out of 201 patients with PCR confirmed SARS-CoV-2 infection, 53 patients were those with severe acute COVID-19 pneumonia and elevated serum markers of COVID-19 severity such as D-Dimer, CRP, and ferritin. Severe patients were predominantly male (n = 44; 83%) and were on average 15 years older than patients with milder COVID-19 (57 years in severe COVID-19 vs. 48 years in mild/moderate COVID-19; P<0.001). Around half of the severe patients (n = 27; 54%) had diabetic mellitus comorbidity. Patient characteristics are listed in Table 1.

### Saliva IL-17 level predicts COVID-19 severity

To assess the possibility of using IL-17 saliva level as a biomarker for COVID-19 severity. First, we measured the circulating IL-17 levels in peripheral blood of recruited COVID-19 patients with different severities. As expected, we found that IL-17 mRNA level in whole blood (Fig 1A, close to 1.5 log fold-change (FC) increase of IL-17 mRNA; P<0.001), and protein level in plasma were significantly elevated in severe COVID-19 cases compared to mild/moderate or asymptomatic COVID-19 cases (Fig 1B, mean 63.4 pg.ml$^{-1}$ in severe vs 28.8 pg.ml$^{-1}$ in mild/moderate COVID-19 cases; P<0.001). Next, we evaluated IL-17 level in saliva samples of these patients and found that, similar to the circulatory IL-17 level, its level was significantly elevated in saliva of severe COVID-19 cases (Fig 1C, mean 138.8 pg.ml$^{-1}$ in severe vs 78.8 pg.ml$^{-1}$ in mild/moderate COVID-19 cases; unadjusted P<0.001). Notably, when adjusting for age, male gender, BMI, DM, and serum markers of COVID-19 severity such as CRP, D-dimer, and ferritin, IL-17 saliva level remained significantly associated with disease severity (Fig 1C and 1D, adjusted P<0.001; and AUC of 0.94 [95% CI, 0.90–0.98]). There were also positive correlations with saliva IL-17 level and serum D-dimer, CRP, and ferritin levels of COVID-19 patients (Fig 1E–1G, P<0.001).

Furthermore, clinical reports have associated COVID-19 severity with elevated blood cytokine levels of TNFα and IL-1β [19, 20]. Therefore, we have then evaluated the levels of TNFα and IL-1β in saliva of COVID-19 patients. Of note, salivary levels of TNFα and IL-1β were elevated in severe COVID-19 cases compared to asymptomatic or mild/moderate cases (unadjusted P<0.001); however, after adjusting for age, male gender, BMI, DM, and serum markers of COVID-19 severity such as CRP, D-dimer, and ferritin, TNFα and IL-1β saliva levels were no longer associated with disease severity. (Fig 1H and 1I for TNFα, adjusted P = 0.212; and AUC of 0.79 [95% CI, 0.70–0.88], and Fig 1J and 1K for IL-1β, adjusted P = 0.382; and AUC of 0.77 [95% CI, 0.68–0.86]).

### Higher salivary IL-17 level associated with higher needs for mechanical ventilation and lower survival

Next, we have assessed the association between IL-17, TNFα and IL-1β levels in saliva of severe COVID-19 cases with the survival outcomes of these patients. Stratifying patients by cytokine levels of high versus low using the cutoffs identified in the severe COVID-19 cases, we found that IL-17, but not TNFα or IL-1β cytokine could predict the need for mechanical ventilation and/or overall survival of patients, based on the first available measurement level after hospital admission (Fig 2A–2F). IL-17 (≥138 pg.ml$^{-1}$) was predictive of the need for mechanical ventilation and/or death by days 29 of admission, after adjusting for age, male gender, BMI, DM, and serum markers of COVID-19 severity such as CRP, D-dimer, and ferritin. (Fig 2A,

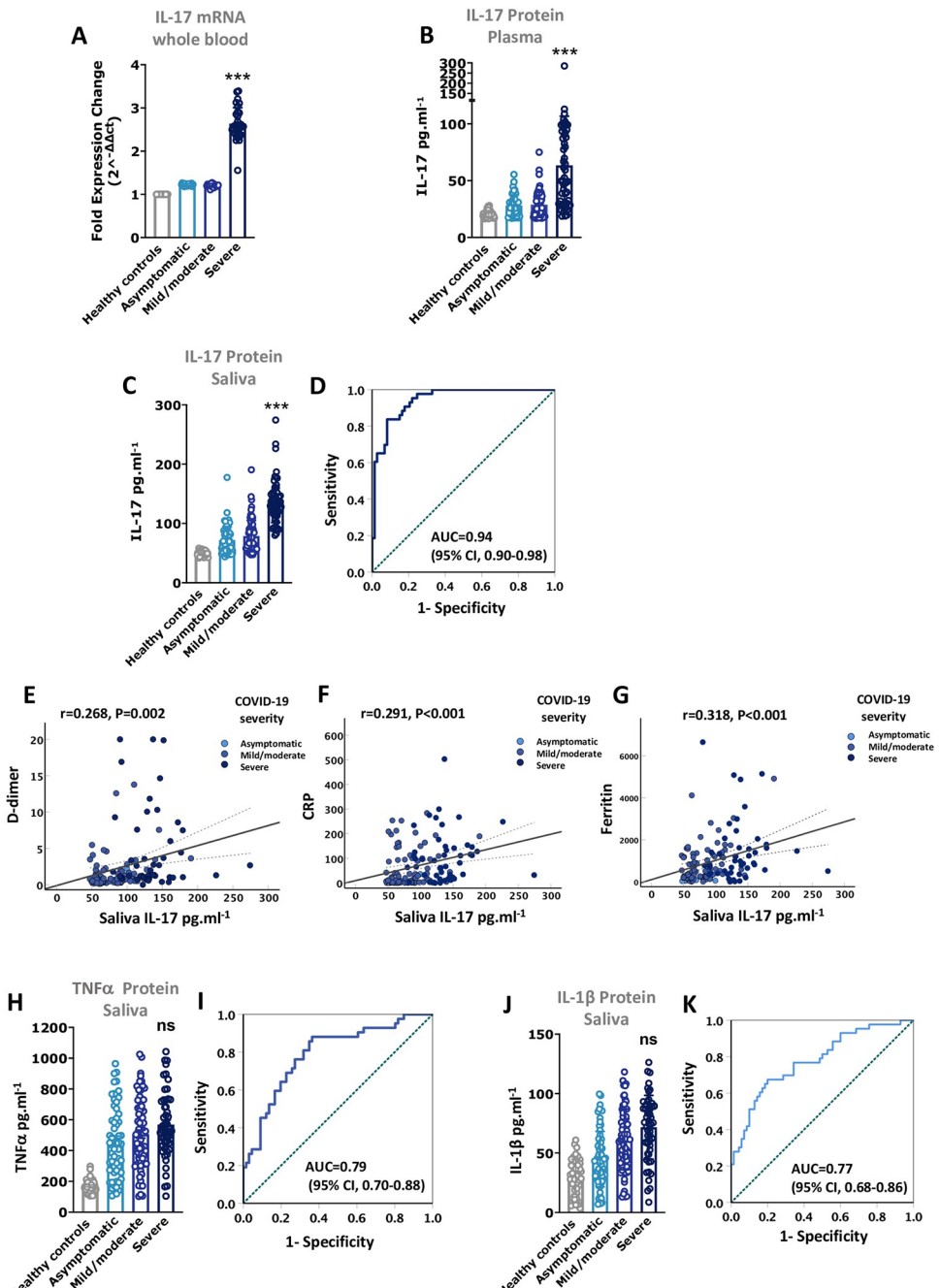

**Fig 1. Higher IL-17 level in saliva of severe COVID-19 patients. (A)** IL-17 mRNA levels in whole blood of COVID-19 patients with different severities. **(B)** IL-17 protein levels in plasma of COVID-19 patients with different severities. **(C and D)** IL-17 protein levels in saliva of COVID-19 patients with different severities and the associated ROC (receiver operating characteristic curve). **(E-G)** Correlation of IL-17 saliva level with serum levels of D-dimer, CRP (C-reactive protein), and ferritin of these patients. **(H-K)** TNFα and IL-1β protein levels in saliva of COVID-19 patients with different severities, and the associated ROCs. Specimens were collected from the following patients with COVID-19 (asymptomatic (n = 67), mild/moderate (n = 81), and severe (n = 53), as well as healthy controls (n = 50). Statistical test: Regression models were adjusted for demographics (age, gender, body mass index), comorbidity (diabetes mellitus) and severity markers of COVID-19 (CRP, D-dimer, and ferritin). ns: Non-significant, * P<0.05, *** P<0.001.

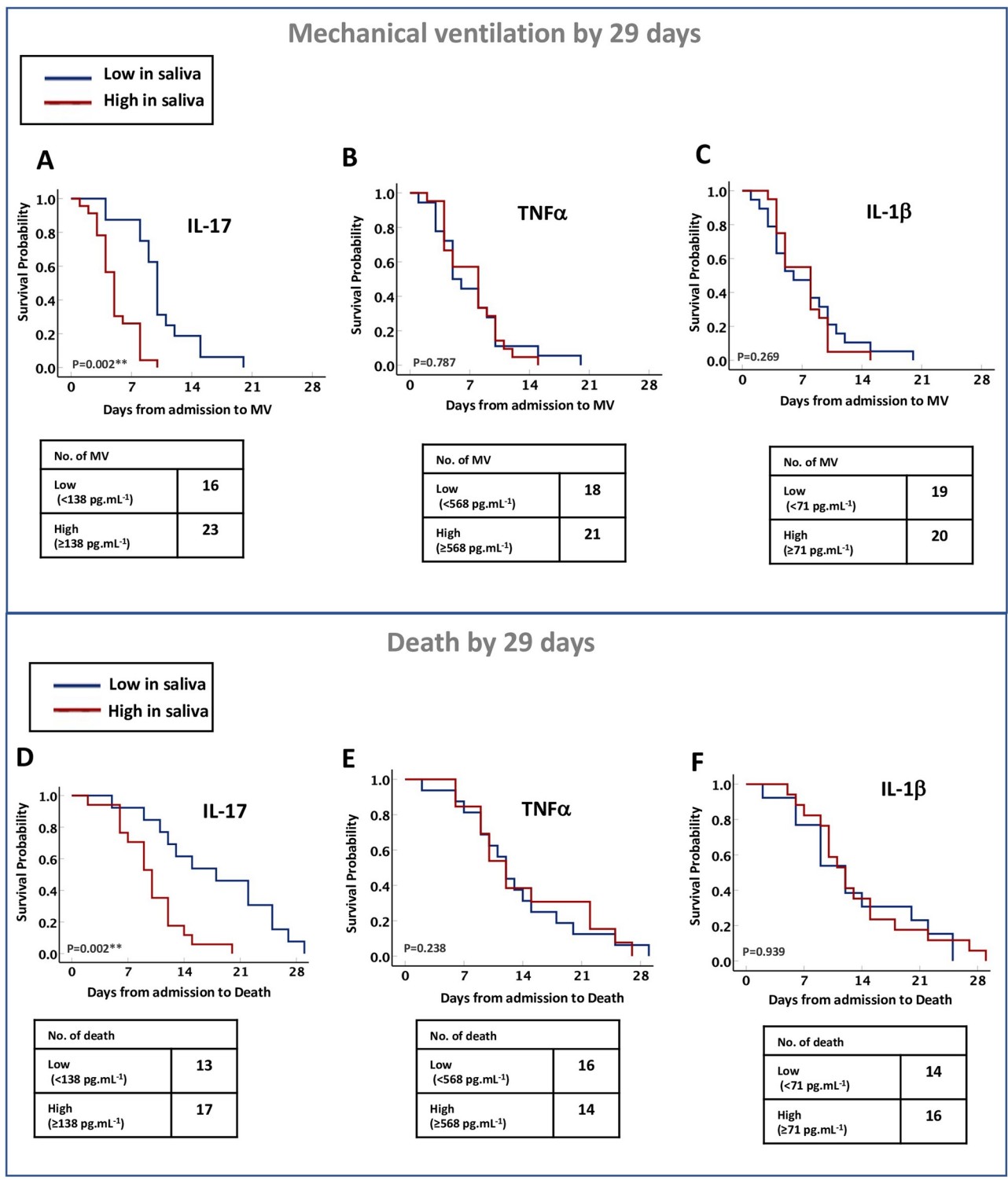

**Fig 2. Increased IL-17 level in saliva of severe COVID-19 patients associated with higher need for mechanical ventilation and/or death by days 29.** Kaplan–Meier survival curves of the need for mechanical ventilation (**A-C**) and/or death (**D-F**), based on the IL-17, TNFα, and IL-1β cytokine levels in saliva of patients with severe COVID-19 (n = 53). Statistical test: Cox proportional models adjusted for patient's demographics factors (age, gender, and body mass index), comorbidities (diabetes mellitus), and COVID-19 related severity serum markers (D-dimer, CRP, and ferritin), with significance indicated by P value of less than 0.05.

adjusted hazard ratio (aHR) for mechanical ventilation, 6.13; 95% CI, 1.9 to 19.1; P = 0.002, and Fig 2D, aHR for death, 11.2; 95% CI, 2.3 to 53.6; P = 0.002).

## Discussion

In the current study, we found that IL-17 level in saliva of COVID-19 patients reflected its circulatory level. Among TNFα or IL-1β saliva levels, higher IL-17 level in saliva of COVID-19 patients was associated with disease severity and worse clinical outcomes, defined as need for mechanical ventilation and/or death within 29 days of admission. This could suggest the potential use of IL-17 as a non-invasive salivary biomarker for COVID-19 severity.

Previously, saliva has been highlighted as a potential non-invasive biological sample for the detection of SARS-CoV-2 [30]. Saliva fluid is a good reservoir for different respiratory viruses. SARS-CoV-2 infects host cells through angiotensin-converting enzyme 2 (ACE2) receptor that is abundantly expressed in salivary gland and oral epithelial cells [31, 32]. In addition to release of viral particles from the infected salivary glands in the oral cavity, viral particles can also be transmitted to saliva from upper and lower respiratory tract [33]. Beside viral particles, important prognostic inflammatory markers including CRP and TNFα were also detected within saliva fluid [34]. Salivary glands are surrounded by rich vessels circulation which facilitate exchange of contents between blood and salivary fluid [35]. Saliva is hypotonic to plasma, and proteins from blood have been shown to enter saliva intracellularly through passive diffusion or active transport, and paracellularly through ultrafiltration at tight junctions between salivary gland cells [36]. Therefore, the elevated level of IL-17 observed in saliva of severe COVID-19 cases could be directly induced by active SARS-CoV-2 infection within the oral cavity and salivary gland, besides the possibly defused protein from blood circulation.

Notably, we found that IL-17 saliva level is a potential biomarker of COVID-19 severity and worse survival outcomes even after adjusting for other risk factors such as patient demographic factors and COVID-19 severity markers. We also measured saliva levels of TNFα and IL-1β, as known markers of inflammation and organ damage, and commonly reported to be elevated in blood of patients with COVID-19 [19, 20, 37]. However, when including additional patients and COVID-19 severity markers, these cytokines were no longer associated with COVID-19 severity and worse clinical outcomes.

Since respiratory tract is the place of SRAS-CoV-2 entry and injury in COVID-19, we evaluated the expression level of IL-17 in gene expression data sets in nasopharyngeal swabs (GSE152075), and lung autopsies from large cohort of patients with COVID-19 (GSE150316). As expected, level of IL-17 was significantly elevated in both nasal swabs as well as lung autopsies of COVID-19 patients compared to those of healthy controls (S1 Fig). IL-17 level during lung inflammation recruit neutrophils, monocytes, and induces production of other proinflammatory cytokines. Higher IL-17 levels in nasopharyngeal swabs and lung autopsies of COVID-19 patients were associated with higher levels of proinflammatory cytokines, including IL-1β, TNFα, IL-6, IL-8, and IL-23 (S2 and S3 Figs). *In vitro*, Th17 cells has the ability to induce the expression of proinflammatory cytokines, including IL-1β, IL-6, IL-8, and TNFα in cell types that are responsive to IL-17, including epithelial cells, fibroblasts, and macrophages [38]. Furthermore, the induction of IL-6, IL-23, IL-1β and TNFα by IL-17 constitutes a positive feedback loop that enhances their production by Th17 cells production and strengthens the effects of IL-17 which may form the basis of a self-sustaining process for IL-17 secretion during infection [39]. In addition, IL-17 signaling can converge with other signaling pathways such as mitogen-activated protein kinase (MAPK), and it can also result in the sequestration of inhibitors of other pathways such as nuclear factor kappa B (NF-κB), and induce their activity [40, 41]. In SARS-CoV-2 infected lung, presence of IL-1β, TNFα, IL-6, and IL-8 indicate IL-17

induced MAPK and NF-κB mediate signaling [11]; MAPK was shown to be activated during the acute phase of SARS-CoV-2 infection in nasopharyngeal swabs of severe COVID-19 patients [42]. Thus, neutralizing IL-17 or its signaling in COVID-19, might constitute an effective strategy in controlling exaggerated uncontrolled lung inflammation following SARS-CoV-2 infection.

In summary, our data suggest a role for IL-17 as a reliable non-invasive salivary biomarker of COVID-19 severity. However, further validation in larger COVID-19 cohorts is needed. Moreover, since our data was drawn from unvaccinated cohort infected with ancestral variants of SARS-CoV-2, further research would be necessary to evaluate level of IL-17 in saliva of vaccinated patients infected with the newly emerged Omicron variants of SARS-CoV-2.

## Supporting information

**S1 Fig. Increased IL-17 gene expression levels in lung and nasopharyngeal swabs of COVID-19 patients.** (A) IL-17 mRNA levels in nasopharyngeal swabs of COVID-19 patients compared to healthy controls (n = 430 COVID-19 patients vs n = 54 healthy controls; GSE152075). (B) IL-17 mRNA levels in lung autopsies of COVID-19 patients compared to healthy controls (n = 17 SARS-CoV-2 infected lung vs. n = 5 healthy lung biopsies; GSE150316). Statistical test: Unpaired t-test or Mann-Whitney U test, depending on the skewness of the data, * $P < 0.05$.
(PDF)

**S2 Fig. Correlation between IL-17 expression level and Th-17 signaling related cytokines/ chemokines such as TNFα, IL-1β, IFNγ, IL-6, neutrophils chemoattractant IL-8, and monocytes chemoattractant, CCL2 in nasopharyngeal swabs of COVID-19 patients (n = 430 COVID-19 patients; GSE152075).** Data show that IL-17 in these COVID-19's nasopharyngeal swabs positively correlate with levels of IL-1β, TNFα, IL-6, IL-8 and CCL2, but not IFNγ. Statistical test: Pearson's coefficient test with two-tailed p-value <0.05 considered significant.
(PDF)

**S3 Fig. Correlation between IL-17 expression level and Th-17 signaling related cytokines/ chemokines such as TNFα, IL-1β, IFNγ, IL-6, IL-8, and CCL2 in lung autopsies of COVID-19 patients (n = 17 SARS-CoV-2 infected lung tissues; GSE150316).** Data show that IL-17 level in COVID-19' lung autopsies positively correlate with levels of TNFα, IL-1β, IFNγ, IL-6, IL-8 and CCL2, but not CCL2. Statistical test: Pearson's coefficient test with two-tailed p-value <0.05 considered significant.
(PDF)

**S1 File. Study raw data.**
(SAV)

## Author Contributions

**Conceptualization:** Fatemeh Saheb Sharif-Askari, Narjes Saheb Sharif-Askari, Rabih Halwani.

**Data curation:** Fatemeh Saheb Sharif-Askari, Shirin Hafezi, Bushra Mdkhana, Hawra Ali Hussain Alsayed, Abdul Wahid Ansari.

**Formal analysis:** Fatemeh Saheb Sharif-Askari, Narjes Saheb Sharif-Askari, Rabih Halwani.

**Funding acquisition:** Rabih Halwani.

**Investigation:** Rabih Halwani.

**Methodology:** Fatemeh Saheb Sharif-Askari, Bassam Mahboub, Rabih Halwani.

**Project administration:** Rabih Halwani.

**Resources:** Qutayba Hamid, Rabih Halwani.

**Supervision:** Rabih Halwani.

**Validation:** Fatemeh Saheb Sharif-Askari, Narjes Saheb Sharif-Askari.

**Visualization:** Fatemeh Saheb Sharif-Askari.

**Writing – original draft:** Fatemeh Saheb Sharif-Askari, Narjes Saheb Sharif-Askari, Shirin Hafezi, Bushra Mdkhana, Hawra Ali Hussain Alsayed, Abdul Wahid Ansari, Bassam Mahboub, Adel M. Zakeri, Mohamad-Hani Temsah, Walid Zahir, Qutayba Hamid, Rabih Halwani.

**Writing – review & editing:** Fatemeh Saheb Sharif-Askari, Narjes Saheb Sharif-Askari, Shirin Hafezi, Bushra Mdkhana, Hawra Ali Hussain Alsayed, Abdul Wahid Ansari, Bassam Mahboub, Adel M. Zakeri, Mohamad-Hani Temsah, Walid Zahir, Qutayba Hamid, Rabih Halwani.

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
