## [Decision Letter · Decision Letter 0]

13 Jul 2022

PONE-D-22-13510Interleukin-17, a salivary biomarker for COVID-19 severityPLOS ONE

Dear Dr. Halwani,

Thank you for submitting your manuscript to PLOS ONE. After careful consideration, we feel that it has merit but does not fully meet PLOS ONE’s publication criteria as it currently stands. Therefore, we invite you to submit a revised version of the manuscript that addresses the points raised during the review process.

We look forward to receiving your revised manuscript.

Kind regards,

Esaki M. Shankar, PhD, FRSB, FRCPath

Academic Editor

PLOS ONE

Journal Requirements:

Additional Editor Comments:

The Manuscript entitled 'Interleukin-17, a salivary biomarker for COVID-19 severity' has attempted to find out the suitable non invasive biomarker that could predict COVID-19 severity. The authors have selected the common cytokines that involve during inflammation "Cytokine storm" and screened in salivary specimen. The following comments can be answered.

Major

IL-17, IL-1β and TNFα were determined in Saliva by commercially available human ELISA kits from Abcam. Whether these kits are standardized for saliva specimen. Can the reviewers have the copy of the kit inserts.

Line 193-195:: "For the COVID-19 nasopharyngeal swabs dataset (GSE152075), the investigators extracted RNA from nasopharyngeal swabs in viral transport media from 430 individuals with SARS-CoV-2 and 54 negative controls. Whether these individuals are different from the 201 adult COVID-19 patients and 10 controls. Please rephrase the manuscript for the clarity of study subjects.

There are only 10 healthy control specimens were used. Whether this is sufficient given a minimum of n=50 individuals in each COVID-19 patients group. Is there a sample size justification to substantiate this.

This study was carried out couple of years ago, whether the outcome is still significant given the introduction of vaccines and reduced COVID-19 hospitalizations.

Minor:

The first paragraph in results (Line 244-247) is just describing the number of individuals in each group in table 1, that can be replaced with the important findings in the table 1 that the authors intend to communicate to the scientific community.

Some references are incomplete such as 1 and 9

The manuscript may need a revision to fix minor punctuation and spelling for Eg:

Line 179: comma before and

Line 194: This can be rephrased - instead of 'investigators extracted RNA', the RNA was extracted

Line 216: non-sever

Reviewers' comments:

Reviewer's Responses to Questions

**Comments to the Author**

1. Is the manuscript technically sound, and do the data support the conclusions?

Reviewer #1: Yes

2. Has the statistical analysis been performed appropriately and rigorously? 

Reviewer #1: Yes

3. Have the authors made all data underlying the findings in their manuscript fully available?

Reviewer #1: Yes

4. Is the manuscript presented in an intelligible fashion and written in standard English?

Reviewer #1: Yes

5. Review Comments to the Author

Reviewer #1: The Manuscript entitled 'Interleukin-17, a salivary biomarker for COVID-19 severity' has attempted to find out the suitable non invasive biomarker that could predict COVID-19 severity. The authors have selected the common cytokines that involve during inflammation "Cytokine storm" and screened in salivary specimen. The following comments can be answered.

Major

IL-17, IL-1β and TNFα were determined in Saliva by commercially available human ELISA kits from Abcam. Whether these kits are standardized for saliva specimen. Can the reviewers have the copy of the kit inserts.

Line 193-195:: "For the COVID-19 nasopharyngeal swabs dataset (GSE152075), the investigators extracted RNA from nasopharyngeal swabs in viral transport media from 430 individuals with SARS-CoV-2 and 54 negative controls. Whether these individuals are different from the 201 adult COVID-19 patients and 10 controls. Please rephrase the manuscript for the clarity of study subjects.

There are only 10 healthy control specimens were used. Whether this is sufficient given a minimum of n=50 individuals in each COVID-19 patients group. Is there a sample size justification to substantiate this.

This study was carried out couple of years ago, whether the outcome is still significant given the introduction of vaccines and reduced COVID-19 hospitalizations.

Minor:

The first paragraph in results (Line 244-247) is just describing the number of individuals in each group in table 1, that can be replaced with the important findings in the table 1 that the authors intend to communicate to the scientific community.

Some references are incomplete such as 1 and 9

The manuscript may need a revision to fix minor punctuation and spelling for Eg:

Line 179: comma before and

Line 194: This can be rephrased - instead of 'investigators extracted RNA', the RNA was extracted

Line 216: non-sever

6. PLOS authors have the option to publish the peer review history of their article (what does this mean?). If published, this will include your full peer review and any attached files.

Reviewer #1: No

---

## [Author Response · Author response to Decision Letter 0]

13 Aug 2022

Date: Jul 13 2022 10:23AM

To: "Rabih Halwani" rhalwani@sharjah.ac.ae

From: "PLOS ONE" plosone@plos.org

Subject: PLOS ONE Decision: Revision required [PONE-D-22-13510]

PONE-D-22-13510

Interleukin-17, a salivary biomarker for COVID-19 severity

PLOS ONE

Dear Dr. Halwani,

Thank you for submitting your manuscript to PLOS ONE. After careful consideration, we feel that it has merit but does not fully meet PLOS ONE’s publication criteria as it currently stands. Therefore, we invite you to submit a revised version of the manuscript that addresses the points raised during the review process.

We look forward to receiving your revised manuscript.

Kind regards,

Esaki M. Shankar, PhD, FRSB, FRCPath

Academic Editor

PLOS ONE

Journal Requirements:

Additional Editor Comments:

We warmly thank the editor, our replies to the below raised points can be found in the “Reviewer #1” section of this letter.

The Manuscript entitled 'Interleukin-17, a salivary biomarker for COVID-19 severity' has attempted to find out the suitable non invasive biomarker that could predict COVID-19 severity. The authors have selected the common cytokines that involve during inflammation "Cytokine storm" and screened in salivary specimen. The following comments can be answered.

Major

IL-17, IL-1β and TNFα were determined in Saliva by commercially available human ELISA kits from Abcam. Whether these kits are standardized for saliva specimen. Can the reviewers have the copy of the kit inserts.

Line 193-195:: "For the COVID-19 nasopharyngeal swabs dataset (GSE152075), the investigators extracted RNA from nasopharyngeal swabs in viral transport media from 430 individuals with SARS-CoV-2 and 54 negative controls. Whether these individuals are different from the 201 adult COVID-19 patients and 10 controls. Please rephrase the manuscript for the clarity of study subjects.

There are only 10 healthy control specimens were used. Whether this is sufficient given a minimum of n=50 individuals in each COVID-19 patients group. Is there a sample size justification to substantiate this.

This study was carried out couple of years ago, whether the outcome is still significant given the introduction of vaccines and reduced COVID-19 hospitalizations.

Minor:

The first paragraph in results (Line 244-247) is just describing the number of individuals in each group in table 1, that can be replaced with the important findings in the table 1 that the authors intend to communicate to the scientific community.

Some references are incomplete such as 1 and 9

The manuscript may need a revision to fix minor punctuation and spelling for Eg:

Line 179: comma before and

Line 194: This can be rephrased - instead of 'investigators extracted RNA', the RNA was extracted

Line 216: non-sever

Reviewers' comments:

Reviewer's Responses to Questions

Comments to the Author

1. Is the manuscript technically sound, and do the data support the conclusions?

Reviewer #1: Yes

2. Has the statistical analysis been performed appropriately and rigorously?

Reviewer #1: Yes

3. Have the authors made all data underlying the findings in their manuscript fully available?

Reviewer #1: Yes

4. Is the manuscript presented in an intelligible fashion and written in standard English?

Reviewer #1: Yes

5. Review Comments to the Author

Reviewer #1: The Manuscript entitled 'Interleukin-17, a salivary biomarker for COVID-19 severity' has attempted to find out the suitable non invasive biomarker that could predict COVID-19 severity. The authors have selected the common cytokines that involve during inflammation "Cytokine storm" and screened in salivary specimen. The following comments can be answered.

Major

IL-17, IL-1β and TNFα were determined in Saliva by commercially available human ELISA kits from Abcam. Whether these kits are standardized for saliva specimen. Can the reviewers have the copy of the kit inserts.

We thank the reviewer for this important comment. 

The following Elisa kits used to measure salivary levels of IL-17, IL-1β, and TNF� were obtained from either R&D systems (IL-17 and IL-1β) or Abcam (TNF�). 

In general, the application of these kits is for measuring cytokines in the cell supernatants, serum, plasma, or other biological fluids with no specific mention of saliva sample. However, these kit advice for optimizing reagent diluent for samples with complex metrics such as plasma and other biologic fluids (see attached insetrs). 

As far as we know there is no existence of Elisa kit (MERCK, Thermo Fisher Scientific, R&D systems, Abcam, and Diaclone) that is standardized for measurement of IL-17, IL-1β, and TNF� cytokines in a saliva specimen. Previously our group [1] and others [2-6] by using these kits have measured levels of different cytokines in saliva samples. However, in our case, to improve assay performance we optimized the diluent for the saliva samples. 

Regarding this we have added the following line to the study method section (pg. 6, lines 240-242), and reads as follows;

“For the assays, saliva samples were centrifuged at 700g for 15 minutes at 4oC, and the supernatant was used. Diluent optimization was performed for the saliva samples. We performed the ELISA assays following the manufacturers' instructions. All samples were measured in duplicates.”

Line 193-195:: "For the COVID-19 nasopharyngeal swabs dataset (GSE152075), the investigators extracted RNA from nasopharyngeal swabs in viral transport media from 430 individuals with SARS-CoV-2 and 54 negative controls. Whether these individuals are different from the 201 adult COVID-19 patients and 10 controls. Please rephrase the manuscript for the clarity of study subjects.

We apologize for the lack of clarity. Cohort of COVID-19 nasopharyngeal swabs dataset (GSE152075) are different from our studied cohort of 201 adult COVID-19 patients. We thus, rephrased the part in the revised version (pg. 6 and 7, lines 325-351) as the following; 

“The gene expression data sets of COVID-19 nasopharyngeal swabs (GSE152075) [7], including 430 patients with SARS-CoV-2 infection and 54 uninfected individuals, and COVID-19 lung autopsies (GSE150316) [8], including 52 COVID-19 fatal cases and 5 SARS-CoV-2-uninfected individuals, were all publicly available at the National Center for Biotechnology Information Gene Expression Omnibus (NCIB GEO, http://www.ncbi.nlm.nih.gov/geo) or the European Bioinformatics Institute (EMBL-EBI, https://www.ebi.ac.uk). 

There are only 10 healthy control specimens were used. Whether this is sufficient given a minimum of n=50 individuals in each COVID-19 patients group. Is there a sample size justification to substantiate this.

We warmly thank the reviewer for this helpful suggestion. In the revised version, we have raised the number of healthy controls to 50 specimens and accordingly updated the manuscript text and Table 1. 

This study was carried out couple of years ago, whether the outcome is still significant given the introduction of vaccines and reduced COVID-19 hospitalizations.

We warmly thank the reviewer for raising this important point. However, since there is no report of IL-17 level in saliva of vaccinated patients with the newly emerged Omicron variants of SARS-CoV-2, we have addressed the issue as future research needs. 

In this regard the discussion parts (pg. 14, lines 627-632) were modified to the following; 

“In summary, our data suggest a role for IL-17 as a reliable non-invasive salivary biomarker of COVID-19 severity. However, further validation in larger COVID-19 cohorts is needed. Moreover, since our data was drawn from unvaccinated cohort infected with ancestral variants of SARS-CoV-2, further research would be necessary to evaluate level of IL-17 in saliva of vaccinated patients infected with the newly emerged Omicron variants of SARS-CoV-2.” 

Minor:

The first paragraph in results (Line 244-247) is just describing the number of individuals in each group in table 1, that can be replaced with the important findings in the table 1 that the authors intend to communicate to the scientific community.

We warmly thank the reviewer for this helpful suggestion. As suggested by the reviewer in the first paragraph of results we have replaced the number of patients in each COVID-19 group with the important patient characterizes mentioned in Table 1. It reads as follows (pg. 9, lines 432-439); 

“Out of 201 patients with PCR confirmed SARS-COV-2 infection, 53 patients were those with severe acute COVID-19 pneumonia and elevated serum markers of COVID-19 severity such as D-Dimer, CRP, and ferritin. Severe patients were predominantly male (n=44; 83%) and were on average 15 years older than patients with milder COVID-19 (57 years in severe COVID-19 vs. 48 years in mild/moderate COVID-19; P<0.001). Around half of the severe patients (n=27; 54%) had diabetic mellitus comorbidity. Patient characteristics are listed in Table 1.”

Some references are incomplete such as 1 and 9

We warmly thank the reviewer for pointing this out. As suggested, we made sure that all the references are complete. Reference 1, for World Health Organization Coronavirus (COVID-19) was updated. 

The manuscript may need a revision to fix minor punctuation and spelling for Eg:

Line 179: comma before and

Line 194: This can be rephrased - instead of 'investigators extracted RNA', the RNA was extracted

Line 216: non-sever

We warmly thank the reviewer for the above comments. As suggested, the manuscript has now been edited for spelling, and overall style. We have also addressed the above requested changes. 

6. PLOS authors have the option to publish the peer review history of their article (what does this mean?). If published, this will include your full peer review and any attached files.

Do you want your identity to be public for this peer review? For information about this choice, including consent withdrawal, please see our Privacy Policy.

Reviewer #1: No

References 

1. Saheb Sharif-Askari F, Saheb Sharif Askari N, Goel S, Mahboub B, Ansari AW, Temsah M-H, et al. Upregulation of IL-19 cytokine during severe asthma: a potential saliva biomarker for asthma severity. ERJ Open Research. 2021:00984-2020. doi: 10.1183/23120541.00984-2020.

2. van Leeuwen SJM, Proctor GB, Potting CMJ, ten Hoopen S, van Groningen LFJ, Bronkhorst EM, et al. Early salivary changes in multiple myeloma patients undergoing autologous HSCT. Oral Dis. 2018;24(6):972-82. doi: https://doi.org/10.1111/odi.12866.

3. Zielińska K, Karczmarek-Borowska B, Kwaśniak K, Czarnik-Kwaśniak J, Ludwin A, Lewandowski B, et al. Salivary IL-17A, IL-17F, and TNF-*α* Are Associated with Disease Advancement in Patients with Oral and Oropharyngeal Cancer. Journal of Immunology Research. 2020;2020:3928504. doi: 10.1155/2020/3928504.

4. Liukkonen J, Gürsoy UK, Pussinen PJ, Suominen AL, Könönen E. Salivary Concentrations of Interleukin (IL)-1β, IL-17A, and IL-23 Vary in Relation to Periodontal Status. J Periodontol. 2016;87(12):1484-91. doi: https://doi.org/10.1902/jop.2016.160146.

5. Xiao F, Du W, Zhu X, Tang Y, Liu L, Huang E, et al. IL-17 drives salivary gland dysfunction via inhibiting TRPC1-mediated calcium movement in Sjögren’s syndrome. Clinical & Translational Immunology. 2021;10(4):e1277. doi: https://doi.org/10.1002/cti2.1277.

6. Techatanawat S, Surarit R, Chairatvit K, Khovidhunkit W, Roytrakul S, Thanakun S, et al. Salivary and serum interleukin-17A and interleukin-18 levels in patients with type 2 diabetes mellitus with and without periodontitis. PLoS One. 2020;15(2):e0228921. doi: 10.1371/journal.pone.0228921.

7. Lieberman NAP, Peddu V, Xie H, Shrestha L, Huang M-L, Mears MC, et al. In vivo antiviral host transcriptional response to SARS-CoV-2 by viral load, sex, and age. PLoS Biol. 2020;18(9):e3000849. doi: 10.1371/journal.pbio.3000849.

8. Desai N, Neyaz A, Szabolcs A, Shih AR, Chen JH, Thapar V, et al. Temporal and spatial heterogeneity of host response to SARS-CoV-2 pulmonary infection. Nature Communications. 2020;11(1):6319. doi: 10.1038/s41467-020-20139-7.

---

## [Editor Report · Decision Letter 1]

6 Sep 2022

Interleukin-17, a salivary biomarker for COVID-19 severity

PONE-D-22-13510R1

Dear Dr. Halwani,

We’re pleased to inform you that your manuscript has been judged scientifically suitable for publication and will be formally accepted for publication once it meets all outstanding technical requirements.

Kind regards,

Esaki M. Shankar, PhD, FRCPath

Academic Editor

PLOS ONE
---

## [Editor Report · Acceptance letter]

13 Sep 2022

PONE-D-22-13510R1 

Interleukin-17, a salivary biomarker for COVID-19 severity 

Dear Dr. Halwani:

I'm pleased to inform you that your manuscript has been deemed suitable for publication in PLOS ONE. Congratulations! Your manuscript is now with our production department. 

Kind regards, 

on behalf of

Dr. Esaki M. Shankar 

Academic Editor

PLOS ONE